# The effects of exercise training on circulating adhesion molecules in adults: A systematic review and meta-analysis

**Mousa Khalafi**[1][º], **Michael E. Symonds**[2][º]*, **Mohammad Hossein Sakhaei**[3][º]*, **Faeghe Ghasemi**[3][º]

**1** Department of Physical Education and Sport Sciences, Faculty of Humanities, University of Kashan, Kashan, Iran, **2** Centre for Perinatal Research, Academic Unit of Population and Lifespan Sciences, School of Medicine, University of Nottingham, Nottingham, United Kingdom, **3** Department of Exercise Physiology, Faculty of Sport Sciences, University of Guilan, Guilan, Iran

º These authors contributed equally to this work.
* michael.symonds@nottingham.ac.uk (MES); sakhaei_phe@yahoo.com (MHS)

**Data Availability Statement:** All relevant data are within the paper and its Supporting Information files.

## Abstract

### Introduction

The current meta-analysis investigated the effects of exercise training on circulating adhesion molecules i.e. soluble intercellular adhesion molecule-1 (sICAM-1) and soluble vascular cell adhesion molecule-1 (sVCAM-1) in adults.

### Method

PubMed, Web of Science, Scopus and Embase were searched to identify original articles, published in English languages journal from inception up to 31 August 2023 that compared the effects of exercise training with non-exercising control on sICAM-1 and sVCAM-1 in adults. Standardized mean differences (SMDs) and 95% CIs were calculated using random-effect models.

### Results

Twenty-three studies including 31 intervention arms and involving 1437 subjects were included in the meta-analysis. Exercise training effectively reduced sICAM-1 [SMD: -0.33 (95% CI -0.56 to -0.11), p = 0.004; $I^2$ = 56.38%, p = 0.001; 23 intervention arms]. Subgroup analyses showed that sICAM-1 decreased in adults with age <60 years (p = 0.01) and BMI $\geq$ 27 kg/m2 (p = 0.002) and those with metabolic disorders (p = 0.004) and cardiovascular diseases (p = 0.005). In addition, aerobic (p = 0.02) and resistance training (p = 0.007) are effective in reducing sICAM-1. However, exercise training did not indicate a superior effect on sVCAM-1 [SMD: -0.12 (95% CI -0.29 to 0.05), p = 0.17; $I^2$ = 36.29%, p = 0.04; 23 intervention arms].

### Conclusion

Our results show that exercise training reduces sICAM-1, but not for sVCAM-1, where both aerobic and resistance training is effective in reducing sICAM-1 in adults with metabolic disorders and cardiovascular diseases.

**Funding:** The author(s) received no specific funding for this work.

**Competing interests:** The authors have declared that no competing interests exist.

## Trial registration

The current meta-analysis was registered at www.crd.york.ac.uk/prospero with ID registration number: CRD42023410474.

## Introduction

Cardiovascular diseases (CVDs) are recognized as the most common cause of mortality internationally [1] with atherosclerosis is one of the severe CVD dysfunction [2]. Atherosclerosis is considered to be a common progressive disease characterized by chronic inflammation and endothelial dysfunction [3]. Circulating inflammatory markers such as interleukin-6 (IL-6) and C-reactive protein (CRP) are closely related with endothelial dysfunction and atherosclerosis [4, 5]. Elevated inflammatory responses can activate the expression of cell adhesion molecules including soluble intercellular adhesion molecule-1 (sICAM-1) and soluble vascular cell adhesion molecule-1 (sVCAM-1) [6, 7]. Adhesion molecules are expressed on leukocytes and endothelium [8] with increased activation of these biomarkers reflecting endothelial activation or damage [7]. Elevated concentrations of sVCAM-1 and sICAM-1 are associated with enhancement of the action of chemokines, which promotes heart disease and arteriosclerosis. In addition, there is evidence that other risk factors, such as obesity and metabolic disorders, are associated with elevated serum adhesion molecules [9, 10]. Arteriosclerosis adhesion molecules are considered a significant risk factor for later cardiometabolic diseases, therefore therapeutic and preventive strategies should have a high priority for identifying subjects with cardiovascular and metabolic risk factors.

Exercise training is an effective non-pharmacological intervention for the prevention and treatment of metabolic and cardiovascular diseases. It can impact on vascular function, visceral fat mass, lipid profiles, inflammatory markers, and glycemia [11–13]. Studies on reducing arteriosclerosis adhesion molecules through exercise training were examined in a 2020 meta-analysis that included patients with coronary artery disease [14]. In addition, a 2014 meta-analysis that included randomized and non-randomized trials, indicated the aerobic exercise decreased the concentration of adhesion molecules [2]. However, the influence of exercise type (resistance, aerobic, and combined), health status (free of diseases vs. with cardiovascular diseases or vs. metabolic disorders) and body mass index (BMI) has not been investigated. Therefore, the current systematic review and meta-analysis investigated the effect of exercise training on sVCAM-1 and sICAM-1 compared with controls. In addition, we undertook subgroup analyses to investigate whether the health status (metabolic disorders, cardiovascular diseases, and free of cardiometabolic diseases), age, BMI, and type of exercise training (aerobic, resistance, or combined) further influenced sVCAM-1 and sICAM-1.

## Methods

### Trial registration

The current meta-analysis was registered (PROSPERO ID registration number: CRD42023410474), and was conducted according to the PRISMA guidelines and the Cochrane Handbook of Systematic Reviews of Interventions.

### Search strategy

An electronic search was conducted in PubMed, Scopus, Web of Science and Embase databases to identify original articles, published until 31 August 2023 using following key words

("exercise" OR "physical activity" OR "exercise training" OR "aerobic training" OR "resistance training" OR "interval training" OR "High-intensity interval training" OR "concurrent training" OR "combined training"))) AND ("inter cellular adhesion molecule*" OR "intercellular adhesion molecule*" OR "ICAM" OR "cell adhesion molecule" OR "CAM" OR "ICAM-1" OR "VCAM-1"). The search was limited to human studies, the English language, and original articles. In addition, the references list of selected studies and Google Scholar were manually searched to ensure that all relevant studies were included in the meta-analysis. In choosing these keywords related to exercise interventions, and adhesion molecules, previous literature was investigated. For linking terms "AND", "OR" operators were used to link synonyms. The search and screening steps were performed by two independent (M Kh and F Gh) researchers. The search strategy is summarized in S1 Table.

## Eligibility criteria and study selection

The criteria for inclusion and exclusion were based on the PIOCS approach which is summarized in Table 1. Our main outcomes were sVCAM-1 and sICAM-1, with independent variables being any type of exercise training including aerobic, resistance, interval and combined training. Additionally, only English language original articles were included with non-original studies, unpublished studies and conference abstracts excluded. All retrieved studies from all databases were exported to Endnote (version 20.2.1) in order to manage and conduct the systematic study selection process. After removing duplicated studies, first step screenings were performed based on abstract, title and keywords against inclusion and exclusion criteria. Subsequently, in the second screening, full-texts were read for all remaining studies to determine their eligibility. All screenings were performed and completed by two independents reviewer (M H S and F G) and any disagreements were resolved by discussion with other reviewers.

## Data extraction and synthesis

Two authors (F Gh and M H S) independently extracted data, and included (1) characteristics of the participants, including sample size, age, BMI, and sex; (2) exercise intervention characteristics, including duration, type, and intensity; (3) outcome variables including circulating ICAM-1 and/or VCAM-1; (4) pre- and post-intervention means, and SDs or mean changes (post values minus pre values) and their SDs. When required, mean and SDs were calculated from medians, ranges, SEs, and/or IQRs. In addition, when required, the Getdata Graph Digitizer software was used to extract means and SDs or mean changes from figures. For studies that included two or more arms of exercise intervention, all were included. Nevertheless, as per the Cochrane guidelines, the number of subjects in the control group was divided.

**Table 1. Inclusion and exclusion criteria based on PIOCS.**

| Category | Inclusion criteria | Exclusion criteria |
| --- | --- | --- |
| Population | Human with ages ≥ 18 years old with no restriction on their sex and health status | Human with ages< 18 years old and trained and athletes participants |
| Intervention | Any type of exercise training with intervention duration ≥ 2 weeks with no restriction on their mode, intensity, frequency and time | Acute exercise and exercise interventions combined with co-interventions such as calorie restriction |
| Comparator | Non-exercise control group | Absence of control group |
| Outcome | At last one measure of circulating ICAM-1 or VCAM-1 using variable method such as ELISA | Measurement of the outcomes from tissues and lake of baseline or post-intervention values |
| study design | Randomized control trial | Single arm trial and non-randomized control trial |

## Quality assessment and sensitivity analyses

Risk of bias was assessed using the Cochrane risk of bias assessment tool within randomized control trials by one reviewer (F Gh) and verified by another (M H S). This tool consists of 7 items: sequence generation, allocation concealment, blinding of participants personnel, blinding of outcomes assessment, incomplete outcomes data, selective outcome reporting and other bias (S2 Table). In addition, to evaluate of the overall quality of the evidence, the Grading of Recommendations, Assessment, Development and Evaluation (GRADE) was used (S3 Table) [15]. GRADE analysis assesses inconsistency, indirectness, impression, risks of bias and other factor at the, in as similar manner to previous reviews [16]. Based on GRADE criteria, the estimated effect of outcomes was very low, low, moderate and high quality [15]. Sensitivity analyses were performed by removing each study individually to determine whether the results were dependent on a single study or not.

## Statistical analyses

To investigate the effects of exercise interventions on sVCAM-1 and sICAM-1, standardized mean differences (SMD) and 95% Cis were calculated using random-effects models. The mean change or pre- and post-intervention scores, and their sample sizes and SDs were used to determine SMD and 95% CIs for each analysis. Since the sVCAM-1 and sICAM-1 values were reported using different units, it was necessary to use SMD. Random-effects models were used because heterogeneity was expected, given methodological and clinical variability, and may have affected the results [17]. Sub-group analyses were performed based on mean BMI (BMI $<27$ kg/m$^2$ vs. BMI $\geq 27$ kg/m$^2$), health status (metabolic disorders, cardiovascular diseases, and free of cardiometabolic diseases), mean age (age $<60$ years vs. age $\geq 60$ years), and training type (aerobic, resistance, and combined training). Heterogeneity was assessed using the I$^2$ statistic, which interprets I$^2$ according to the Cochrane guidelines as follows: 25%, 50%, and 75% indicate low, moderate, and high heterogeneity, respectively. Additionally, publication bias was assessed using visual interpretation of funnel plots with Egger's tests. Significance for effect size and heterogeneity was considered at $p<0.05$, and for Egger's tests at $p<0.1$.

# Results

## Search results

Our initial search strategy revealed 1239 records from PubMed, 1696 records from Scopus, 892 records from Web of Science and 1024 records from Embase. After removing duplicates and screening titles and abstracts, 78 articles were identified for full-paper analysis based on inclusion/exclusion criteria. Finally, 55 studies were excluded, and 23 studies involving 31 intervention arms met all eligibility criteria. They were included in the meta-analysis, of which four studies assessed sICAM-1 [18–21], seven studies assessed sVCAM-1 [22–28], and 12 studies assessed sICAM-1and sVCAM-1 [3, 8, 7, 29–37] (Fig 1).

## Participant characteristics

A total of 1437 adults were included, with the range of sample sizes being 26 [26] to 140 [8]. The mean age of participants varied from 24 [22] to 70 years [23], and the mean BMI of participants was 22 [26] to 34 kg/m$^2$ [22]. Four studies included only males [20, 25, 29, 37], and six only females [19, 22, 26, 30, 32, 33], whereas 13 included females and males [3, 8, 7, 18, 21, 23, 24, 27, 28, 31, 34–36, 38]. In the meta-analysis, participants had a wide range of health and chronic diseases, including myocardial infarction, type 2 diabetes, percutaneous coronary intervention, syndrome X, coronary artery disease, chronic heart failure, colon cancer

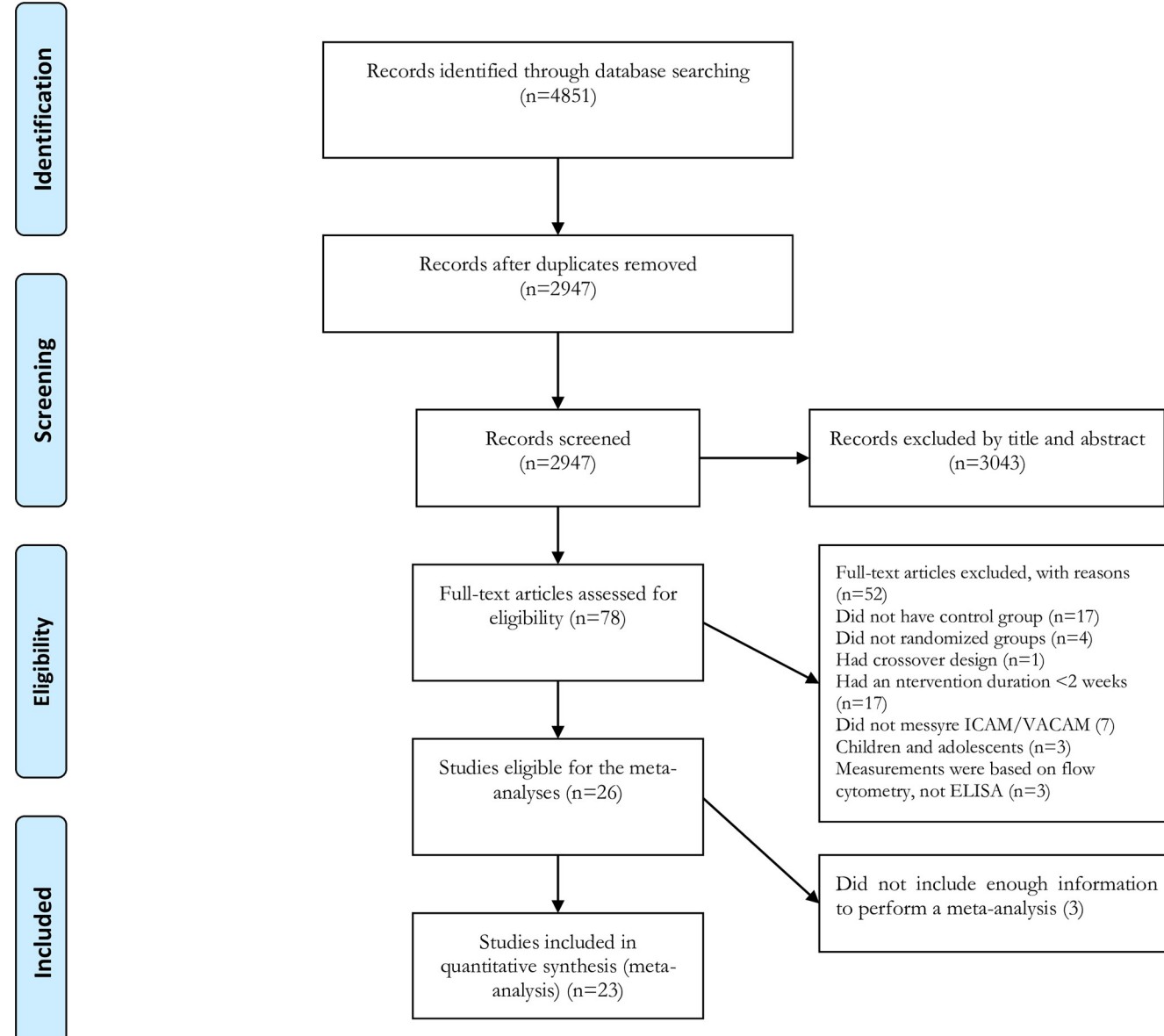

**Fig 1. Flow diagram of systematic literature search.**

survivors, obesity, hypertension, and peripheral artery disease with intermittent claudication (Table 2). In addition, in subgroup analyses, type 2 diabetes, syndrome X, hypertension, and obesity were included together as metabolic disorders [3, 8, 20, 22, 25, 30, 32,], and myocardial infarction, peripheral artery disease with intermittent claudication, coronary artery disease, and chronic heart failure were included as CVD [7, 18, 24, 27, 28, 31, 35, 37]. If participants had no chronic disorders, they were included as free of CVD [19, 21, 23, 26, 29, 33]. Also, one study included both participants with type 2 diabetes and coronary artery disease [34], and another study included participants with colon cancer [36], which were not included in the subgroup analysis.

### Intervention characteristics

The intervention characteristics are presented in Table 2, and studies were included that, compared the effects of exercise training compared with a control group. Intervention duration ranged from two weeks [18] to twelve months [32, 34], whereas the frequency of exercise sessions ranged from two [37] to twelve sessions [18] per week. For exercise training interventions, aerobic training [3, 7, 18, 21, 24, 25, 27, 29, 31, 33, 35–37], resistance training [23, 30, 32], and combined training [8, 22, 26, 34] were the most common. Moreover, in other studies, aerobic, resistance, and combined training [19], aerobic, and resistance training [20, 28] were used as separate exercise arms.

### Meta-analysis

**sICAM-1.**  Based on 23 intervention arms from 16 studies, exercise training decreases sICAM-1 [SMD: -0.33 (95% CI -0.56 to -0.11), p = 0.004] (Fig 2). There was significant heterogeneity amongst studies ($I^2$ = 56.38%, p = 0.001). Both visual interpretation of funnel plots and Egger's test results (p = 0.001) suggested publication bias. The trim and fill method identified seven missing studies from the right side of the plots [SMD: -0.11 (95% CI -0.35 to 0.11)]. Subgroup analyses showed that sICAM-1 decreased in adults <60 years (SMD: -0.36, p = 0.01), and participants with BMI $\geq$ 27 kg/m$^2$ (SMD: -0.68, p = 0.002), and aerobic training (SMD: -0.31, p = 0.02), and resistance training (SMD: -0.80, p = 0.007). For health status, sICAM-1 decreased in participants with metabolic disorders (SMD: -0.59, p = 0.004), and cardiovascular diseases (SMD: -0.39, p = 0.005).

**sVCAM-1.**  Based on 23 intervention arms from 19 studies, exercise training showed no significant decrease in sVCAM-1 [SMD: -0.12 (95% CI -0.29 to 0.05), p = 0.17] when compared with controls (Fig 3). There was significant heterogeneity amongst included studies ($I^2$ = 36.29%, p = 0.04). Both visual interpretation of funnel plots and Egger's test results (p = 0.005) suggested publication bias. The trim and fill method identified eight missing studies from the right side of the plots [SMD: 0.09 (95% CI -0.09 to 0.29)].

**Quality assessment and sensitivity of analyses.**  The methodological quality of individual studies is provided in S2 Table. In addition, based on GRADE criteria, the effect estimates for sICAM-1 and sVCAM-1 were categorized as high and low quality, respectively (S3 Table). Sensitivity analysis showed that omitting individual studies did not change the direction or significance of the results.

## Discussion

Our meta-analysis suggests that exercise training reduces sICAM-1, but not for sVCAM-1, and sICAM-1 was lowered following resistance and aerobic training. In addition, sICAM-1 decreased in individuals with both metabolic disorders and cardiovascular diseases, individuals with obesity, as well as in middle-aged adults.

**Table 2. Summary of demographic characteristics of participants and interventions included in study.**

| First author, year | Sample size (sex) | Health status | Mean age (years±SD) | Mean BMI (kg/m²) | Design | Outcomes | Exercise Mode | Exercise protocol | Control protocol | Session per week | Intervention duration |
|---|---|---|---|---|---|---|---|---|---|---|---|
| Aksoy et al, 2015 [7] | 57 (F & M) | Chronic heart failure | EX₁:63.7±8.8 EX₂:59.6±6.9 CON:57.5±11.2 | EX₁:28.4±4.9 EX₂:30.1±5.1 CON:29.1±4.2 | RCT | sICAM-1 sVCAM-1 | Aerobic interval | $AT_1$;30-min at 50–75% of $VO2_{max}$ by ergometer $AT_2$: 9 sets of 60-s at a determined intensity by 30-s at 30 W by cycling | nonexercising | 3 | 10 weeks |
| Andrade-Lima et al, 2021 [37] | 35 (M) | Peripheral artery disease and Intermittent claudication | EX:66±8.0 CON:69±12.0 | EX:25.7±3.2 CON:25.4±3.6 | RCT | sICAM-1 sVCAM-1 | Aerobic interval | 60-min with 15 bouts of 2-min walking on a treadmill at an intensity corresponding to the heart rate obtained at the pain threshold interspersed by 2-min of upright rest | 30-min of stretching for all body segments | 2 | 12 weeks |
| Barone Gibbs et al, 2012 [8] | 140 (F & M) | Type 2 diabetes | EX:58.0±5.0 CON:56.0±6.0 | EX:32.3±5.3 CON:33.5±4.3 | RCT | sICAM-1 sVCAM-1 | Combined | AT:45-min at 60–90% of $HR_{max}$ RT:whole body exercises; 7 exercises, 2 sets with 12–15 reps at 50% of 1RM | received a usual care | 3 | 6 months |
| Boeno et al, 2020 [28] | 42 (F & M) | Hypertensive patients | EX₁:45.8±6.8 EX₂:46.1±7.2 CON:44.3±8.3 | EX₁:32.9±4.5 EX₂:31.5±4.5 CON:35.0±3.1 | RCT | sVCAM-1 | Aerobic&resistance | AT:45–50 min at 60–80% of HRR by treadmill RT: whole body exercises; 7 exercises, 2–3 sets with 8–20 reps at 50% of 1RM | maintain usual physical activity and nutritional habits | 3 | 12 weeks |
| Brown et al, 2018 [36] | 39 (F & M) | Colon Cancer Survivors X | EX₁:58.2±9.8 EX₂:53.1±10.5 CON:57.9±9.7 | EX₁:29.5±4.3 EX₂:32.4±6.9 CON:29.2±6.0 | RCT | sICAM-1 sVCAM-1 | Aerobic | $AT_1$;150-min per week at 50–70% of $HR_{max}$ by in-home treadmills $AT_2$;300-min per week at 50–70% of $HR_{max}$ by in-home treadmills | received a usual care | 3–5 | 6 months |
| Byrkjeland et al, 2011 [35] | 80 (F & M) | Chronic heart failure | EX:68.8±7.9 CON:71.5±7.8 | ND | RCT | sICAM-1 sVCAM-1 | Aerobic | 50-min including three intervals of high intensity at 15–18 BRPE and two periods of moderate intensity at 11–13 BRPE by group-based simple aerobic exercise | was not discouraged from regular physical activity | 2 | 4 months |
| Byrkjeland et al, 2017 [34] | 137 (F & M) | Type 2 diabetes and coronary artery disease | EX:64.6±7.6 CON:63.2±7.2 | EX:29.1±4.0 CON:29.0±5.6 | RCT | sICAM-1 sVCAM-1 | Combined | 60-min with BRPE≥15 twice a week consisted of group based exercisea and third weekly session of home-based individual exercise, approximately two-third was aerobic and one-third resistance exercises | were not discouraged from regular physical activity or exercise | 3 | 12 months |
| Castells-Sanchez et al, 2022 [21] | 48 (F & M) | Healthy | EX:58.4±5.1 CON:56.6±6.0 | EX:28.1±5.5 CON:30.5±5.7 | RCT | sICAM-1 | Aerobic | 30–45 min at 9–14 BRPE by brisk walking | were not alter their regular lifestyle | 5 | 12 weeks |

*(Continued)*

**Table 2.** (Continued)

| First author, year | Sample size (sex) | Health status | Mean age (years±SD) | Mean BMI (kg/m²) | Design | Outcomes | Exercise Mode | Exercise protocol | Control protocol | Session per week | Intervention duration |
|---|---|---|---|---|---|---|---|---|---|---|---|
| Connolly et al, 2016 [33] | 62 (F) | Premenopausal | EX$_1$:44.0±5.0 EX$_2$:46.0±4.0 CON:45.0±4.0 | EX$_1$:>25 EX$_2$:>25 CON:>25 | RCT | sICAM-1 sVCAM-1 | HIIT & Aerobic | HIIT:6 to 10 reps of 30-s at all-out swimming by 2-min of passive recovery AT: 60-min at low intensity consisted of continuous front-crawl swimming | no training or lifestyle changes in the same period | 3 | 15 weeks |
| Fernandes et al, 2011 [27] | 34 (F & M) | Coronary artery disease | EX:60.7±6.7 CON:59.5±7.3 | EX:28.6 ±5.9 CON:27.6 ±3.6 | RCT | sVCAM-1 | Aerobic | 40 min of cycling at a target heart rate between anaerobic threshold and respiratory compensation point | received recommendations for lifestyle modification | 3 | 4 months |
| Hyun-Hun et al, 2019 [26] | 26 (F) | Elderly | EX:69.6±2.1 CON:69.6 ±2.2 | EX:23.2 ±1.5 CON:21.1 ±0.8 | RCT | sVCAM-1 | Combined | AT: 30-min, intensity of according to the individuals' level of exercise by walking on a treadmill RT: 50-min using 0.5~2kg dumbbell and thera band | habitual daily life activities | 4 | 12 weeks |
| Koh et al, 2018 [3] | 27 (F & M) | Obesity | EX:30.0 ±16.0 CON:25.0 ±8.0 | EX:33.24 ±6.1 CON:32.5 ±5.7 | RCT | sICAM-1 sVCAM-1 | Aerobic | 60-min at 70% of HR$_{max}$ by treadmill | not exercise | 3 | 4 weeks |
| Lim et al, 2015 [25] | 30 (M) | Syndrome X | EX:56.8±1.8 CON:58.3 ±1.9 | EX:23.4 ±1.4 CON:23.3 ±2.3 | RCT | sVCAM-1 | Aerobic | 30-min with 3 sets of 10-min at 60~79% of HR$_{max}$ separated by intervals of ≥4 hours by brisk walking | control group | 3 | 10 weeks |
| Munk et al, 2011 [24] | 40 (F & M) | Percutaneous coronary intervention | EX:59.5 ±10.0 CON:60.7 ±9.0 | EX:26.1 ±4.0 CON:28.4 ±3.3 | RCT | sVCAM-1 | HIIT | 4 sets 4-min at 80~90% of HR$_{max}$ by 3-min active recovery at 60~70% of HR$_{max}$ | not exercise | 3 | 6 months |
| Nikseresht et al, 2014 [20] | 33 (M) | Obesity | EX$_1$:40.4 ±5.2 EX$_2$:39.6 ±3.7 CON:38.9 ±4.1 | ND | RCT | sICAM-1 | Resistance& HIIT | RT:40–65 min, whole body exercises; 5–11 exercises, 1–4 sets with 2–20 reps at 40~95% of 1RM by nonlinear resistance training HIIT:4 sets 4-min at 80~90% of HR$_{max}$ by 3-min recovery at 80~90% of HR$_{max}$ by running on a treadmill | maintained a sedentary lifestyle | 3 | 12 weeks |
| Olson et al, 2007 [32] | 32 (F) | Obesity | EX:39.0±5.0 CON:38.0 ±6.0 | EX:26.9 ±3.0 CON:27.0 ±3.0 | RCT | sICAM-1 sVCAM-1 | Resistance | whole body exercises; 2–3 sets with 6–12 reps by isotonic variable resistance machines and free weights | continue their usual activities | 2 | 1 year |
| Ribeiro et al, 2012 [31] | 42 (F & M) | Myocardial infarction | EX:54.3 ±10.8 CON:57.0 ±7.6 | EX:28.4 ±4.0 CON:26.6 ±4.6 | RCT | sICAM-1 sVCAM-1 | Aerobic | 35-min at 65~70% of HR$_{max}$ by cycloergometer or treadmill | provided usual medical care | 3 | 8 weeks |

*(Continued)*

**Table 2.** (Continued)

| First author, year | Sample size (sex) | Health status | Mean age (years±SD) | Mean BMI (kg/m²) | Design | Outcomes | Exercise Mode | Exercise protocol | Control protocol | Session per week | Intervention duration |
|---|---|---|---|---|---|---|---|---|---|---|---|
| Rosety et al, 2016 [30] | 48 (F) | Obesity | EX:67.3±2.1 CON:68.1±2.3 | EX:31.2±1.0 CON:31.6±1.2 | RCT | sICAM-1 sVCAM-1 | Resistance | whole body exercises; 6 stations at 8 RM by circuit resistance training | not take part in any training program | 3 | 12 weeks |
| Sjogren et al, 2010 [29] | 79 (M) | Healthy | EX:46.0±6.0 CON:47.0±4.0 | EX:25.3±2.9 CON:24.5±3.0 | RCT | sICAM-1 sVCAM-1 | Aerobic | 30–45 min at 60–80% of $HR_{max}$ by regular physical activities of an aerobic | maintain their previous lifestyles | 2–3 | 6 months |
| Soori et al, 2017 [19] | 32 (F) | Postmenopausal | $EX_1$:45.0–60.0 $EX_2$:45.0–60.0 $EX_3$:45.0–60.0 CON:45.0–60.0 | $EX_1$:31.0 ±1.8 $EX_2$:30.1 ±0.4 $EX_3$:30.7 ±1.1 CON:30.7 ±1.0 | RCT | sICAM-1 | Aerobic& resistance& combined | AT: 45-min at 40% of $HR_{max}$ by swimming or walking in the water RT:45-min, whole body exercises; 6 exercise, 3 sets with 10–12 reps at 40–60% of 1RM CT:22-min RT, 2 sets with 10–12 reps at 40% of 1RM following 22-min AT, at 40% of $HR_{max}$ | no regular physical activity | 3 | 10 weeks |
| Timon et al, 2021 [23] | 41 (F & M) | Older adults | EX:70.3±3.3 CON:70.5 ±4.0 | EX:27.1 ±3.9 CON:26.8 ±2.6 | RCT | sVCAM-1 | Resistance | 30-min, whole body exercises; 9 exercise, 3 sets with 12–15 reps at 6–8 RPE and 6 exercise with elastic bands and 2 exercise with kettlebells, 4–10 kg and 3 times, 15-30s front plank | instructed to continue with their normal daily activities | 3 | 24 weeks |
| Vasic et al, 2019 [18] | 90 (F & M) | Patients with a recent myocardial infarction | $EX_1$:62.4 ±7.6 $EX_2$:56.7 ±8.4 CON:60.6 ±8.3 | $EX_1$:29.7 ±5.5 $EX_2$:29.9 ±4.3 CON:29 ±3.2 | RCT | sICAM-1 | Aerobic + calisthenics | $AT_1$:30 min at 60–80% $HR_{peak}$ by land-based exercise training plus 30 min at 60–80% $HR_{peak}$ by calisthenics exercise training on land $AT_2$:30 min at 60–80% $HR_{peak}$ by water-based exercise training plus 30 min at 60–80% $HR_{peak}$ by calisthenics exercise training in thermo-neutral water | advised physical activity at home while waiting | 12 | 2 weeks |
| Woudberg et al, 2018 [22] | 45 (F) | Obesity | EX:22.8±3.1 CON:24.5 ±3.5 | EX:34.4 ±2.7 CON:33.3 ±3.1 | RCT | sVCAM-1 | Combined | AT: 40–60 min dancing, running, skipping, and stepping at 75–80% of $HR_{peak}$ RT: whole body exercises; using body weight, bands and free weights | no exercise | 4 | 12 weeks |

**Abbreviations:** *Ex* exercise intervention; *CON* control; *F* female, *M* male; *HIIT* high-intensity interval training; $VO_{2max/peak}$ maximal or peak oxygen uptake; $HR_{max/peak}$ maximal or peak heart rate; *HRR* heart rate reverse; *RT* resistance; *AT* aerobic; *CT* combined; *AI* aerobic interval; *reps* repetitions; *1RM* one-repetition maximum; *BRPE* borg rating of perceived exertion; *sICAM-1*soluble intercellular adhesion molecule 1; *sVCAM-1*soluble vascular adhesion molecule 1; *ND* not-described

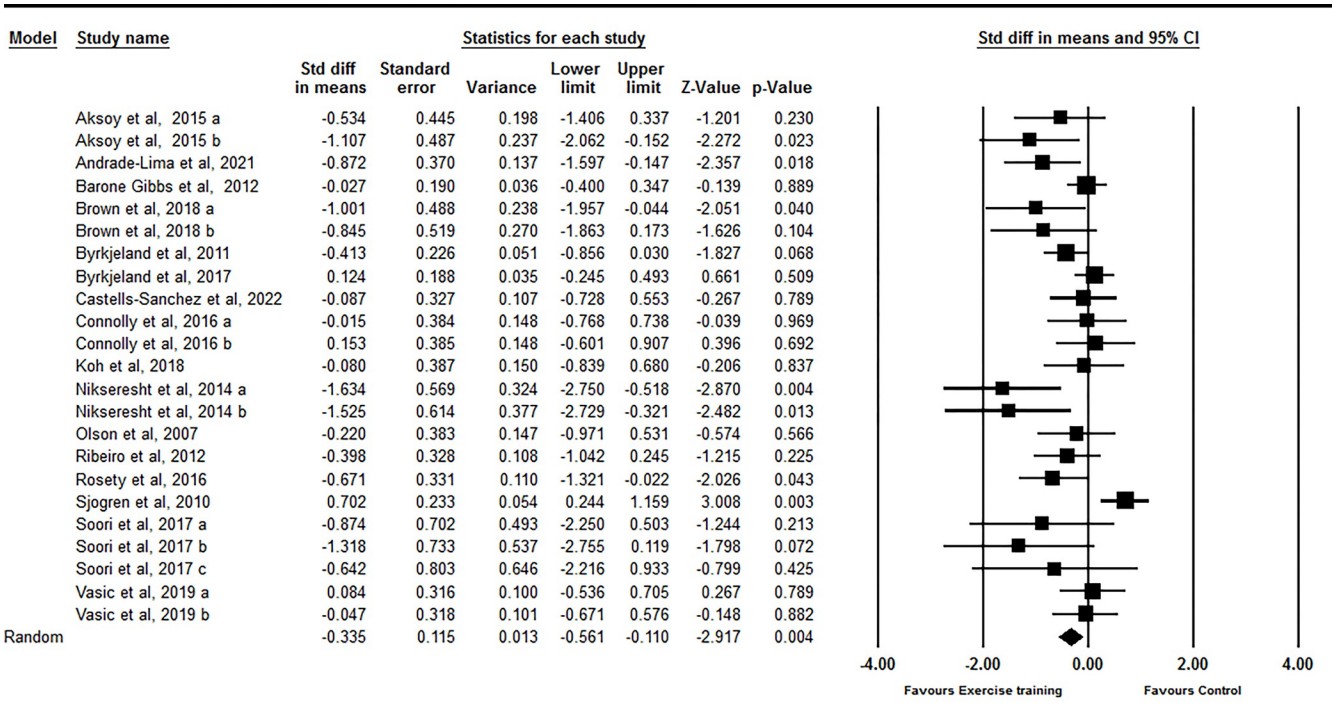

**Fig 2. Forest plot of the effects of exercise training versus control on sICAM-1.** Data are reported as SMD (95% confidence limits). SMD: standardized mean difference.

The importance of exercise training on sICAM-1 is emphasized by its role in cancer, immune syndromes, CVD, and chronic inflammation [39, 40]. Systematic review studies have suggested that exercise training is effective for reducing sICAM-1 [2] and is practical for non-significant reducing sICAM-1 in individuals with coronary artery disease [14]. However, some included studies used exercise program alongside a comprehensive cardiac rehabilitation intervention, which limits imputing these results to an absolute effect of exercise [14]. In addition, in a 2018 systematic review, has not been positive effect of exercise interventions on sICAM-1 reduction in heart failure patients [41]. The potential mechanisms underlying the decrease in sICAM-1 may be explained by greater nitric oxide bioavailability, and improving endothelial function [42, 43].

However, the type of exercise may play an important role in decreasing sICAM-1. When investigating subgroup analyses, we found that aerobic and resistance interventions, relative to combined intervention, are effective for reducing sICAM-1. Aerobic interventions lead to an increase in blood flow and shear stress, followed by enhanced nitric oxide bioavailability [12, 44]. A combination of aerobic, and resistance training can improve body composition and lipid profile, which may enable reduced adhesion molecules and improved endothelial function [45–49]. More importantly, inflammatory stimuli are the main stimulus for adhesion molecules on vascular endothelium and circulating leukocytes [50]. Exercise may enhance the number of vascular endothelial progenitor cells [51, 52], thus improving the regeneration of endothelial cells after vascular injury and reduce chronic inflammation [53]. Accordingly, aerobic training can reduce sICAM-1 through the improvement of inflammatory factors, that may explain why our subgroup analysis by health status, (metabolic disorders, with and without CVD), showed that disease was accompanied with reduced sICAM-1.

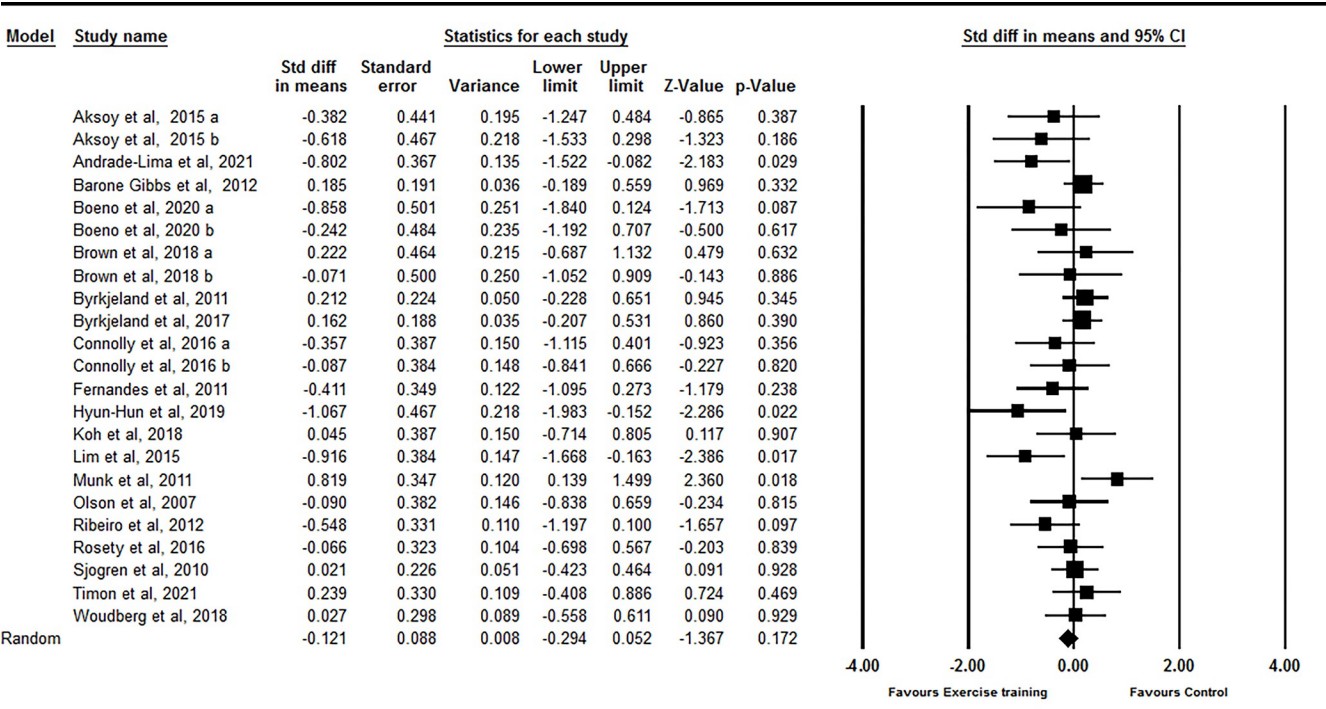

| Model | Study name | | Statistics for each study | | | | | | | Std diff in means and 95% CI |
|---|---|---|---|---|---|---|---|---|---|---|
| | | Std diff in means | Standard error | Variance | Lower limit | Upper limit | Z-Value | p-Value | |
| | Aksoy et al, 2015 a | -0.382 | 0.441 | 0.195 | -1.247 | 0.484 | -0.865 | 0.387 | |
| | Aksoy et al, 2015 b | -0.618 | 0.467 | 0.218 | -1.533 | 0.298 | -1.323 | 0.186 | |
| | Andrade-Lima et al, 2021 | -0.802 | 0.367 | 0.135 | -1.522 | -0.082 | -2.183 | 0.029 | |
| | Barone Gibbs et al, 2012 | 0.185 | 0.191 | 0.036 | -0.189 | 0.559 | 0.969 | 0.332 | |
| | Boeno et al, 2020 a | -0.858 | 0.501 | 0.251 | -1.840 | 0.124 | -1.713 | 0.087 | |
| | Boeno et al, 2020 b | -0.242 | 0.484 | 0.235 | -1.192 | 0.707 | -0.500 | 0.617 | |
| | Brown et al, 2018 a | 0.222 | 0.464 | 0.215 | -0.687 | 1.132 | 0.479 | 0.632 | |
| | Brown et al, 2018 b | -0.071 | 0.500 | 0.250 | -1.052 | 0.909 | -0.143 | 0.886 | |
| | Byrkjeland et al, 2011 | 0.212 | 0.224 | 0.050 | -0.228 | 0.651 | 0.945 | 0.345 | |
| | Byrkjeland et al, 2017 | 0.162 | 0.188 | 0.035 | -0.207 | 0.531 | 0.860 | 0.390 | |
| | Connolly et al, 2016 a | -0.357 | 0.387 | 0.150 | -1.115 | 0.401 | -0.923 | 0.356 | |
| | Connolly et al, 2016 b | -0.087 | 0.384 | 0.148 | -0.841 | 0.666 | -0.227 | 0.820 | |
| | Fernandes et al, 2011 | -0.411 | 0.349 | 0.122 | -1.095 | 0.273 | -1.179 | 0.238 | |
| | Hyun-Hun et al, 2019 | -1.067 | 0.467 | 0.218 | -1.983 | -0.152 | -2.286 | 0.022 | |
| | Koh et al, 2018 | 0.045 | 0.387 | 0.150 | -0.714 | 0.805 | 0.117 | 0.907 | |
| | Lim et al, 2015 | -0.916 | 0.384 | 0.147 | -1.668 | -0.163 | -2.386 | 0.017 | |
| | Munk et al, 2011 | 0.819 | 0.347 | 0.120 | 0.139 | 1.499 | 2.360 | 0.018 | |
| | Olson et al, 2007 | -0.090 | 0.382 | 0.146 | -0.838 | 0.659 | -0.234 | 0.815 | |
| | Ribeiro et al, 2012 | -0.548 | 0.331 | 0.110 | -1.197 | 0.100 | -1.657 | 0.097 | |
| | Rosety et al, 2016 | -0.066 | 0.323 | 0.104 | -0.698 | 0.567 | -0.203 | 0.839 | |
| | Sjogren et al, 2010 | 0.021 | 0.226 | 0.051 | -0.423 | 0.464 | 0.091 | 0.928 | |
| | Timon et al, 2021 | 0.239 | 0.330 | 0.109 | -0.408 | 0.886 | 0.724 | 0.469 | |
| | Woudberg et al, 2018 | 0.027 | 0.298 | 0.089 | -0.558 | 0.611 | 0.090 | 0.929 | |
| Random | | -0.121 | 0.088 | 0.008 | -0.294 | 0.052 | -1.367 | 0.172 | |

**Fig 3. Forest plot of the effects of resistance training versus control on sVCAM-1.** Data are reported as SMD (95% confidence limits). SMD: standardized mean difference.

In addition, we showed the effectiveness of exercise training on reducing sICAM-1 in subjects with high BMI which may be related to higher levels of adhesion molecules [53–56]. In this regard, Ribeiro suggested that a positive effect of exercise is more apparent when the baseline levels of inflammatory markers and cytokines are higher [31]. Thus, it is likely that sICAM-1 reduction rate in response to exercise interventions is directly related to the basal levels of these molecules. Although, our results align with the aforementioned hypothesis, additional studies are needed.

The current meta-analysis failed to find a significant reduction in sVCAM-1 which is disagree with the previous systematic reviews [2, 57]. However, our results agree with several systematic reviews that exercise training failed to provide strong evidence for reducing sVCAM-1 in patients with heart failure [41] coronary artery disease [14]. It seems that exercise intensity paly important role in reducing sVCAM-1. High-intensity training is related to a larger reduction of sVCAM-1, as compared with moderate training intensity [10]. However, only two of our studies [24, 33] used a high-intensity exercise intervention, and most studies used moderate-intensity exercise. Further studies are still needed to clear the effect of exercise intensity on reducing sVCAM-1.

The study had several imitations that should be considered when interpreting our results. There was significant heterogeneities amongst studies that based on our subgroup analyses are probably due to health status and ages of participants and type of exercise intervention. Although, other components, such as duration and intensity of exercise and sample size may also contribute. We observed a significant publication bias for several of the analyses that were corrected for, using the trim-and-fill correction method. In addition, the quality of included studies was demonstrated with high risk of bias and sVCAM-1 classified as low quality rating

identified through GRADE analysis due to it being an indirect outcome with a high risks of bias.

## Conclusion

Despite the lack of effect on sVCAM-1, exercise training appears to be an effective intervention for reducing sICAM-1 in adults. However, health status, age and BMI of participants may be important moderators where such an adaptation could be enhanced in those who are obese and middle-aged, as well as suffering from metabolic disorders and cardiovascular diseases.

## Supporting information

**S1 Checklist. PRISMA 2020 checklist.**
(DOCX)

**S1 Table. Search strategy.**
(DOCX)

**S2 Table. Risk of bias assessment.**
(DOCX)

**S3 Table. GRADE analysis of the overall quality of the evidence.**
(DOCX)

## Author Contributions

**Conceptualization:** Mousa Khalafi, Michael E. Symonds, Mohammad Hossein Sakhaei, Faeghe Ghasemi.

**Data curation:** Mousa Khalafi, Mohammad Hossein Sakhaei, Faeghe Ghasemi.

**Formal analysis:** Mousa Khalafi, Mohammad Hossein Sakhaei.

**Investigation:** Mousa Khalafi, Michael E. Symonds, Mohammad Hossein Sakhaei, Faeghe Ghasemi.

**Methodology:** Mousa Khalafi, Michael E. Symonds, Mohammad Hossein Sakhaei, Faeghe Ghasemi.

**Software:** Mousa Khalafi.

**Validation:** Michael E. Symonds, Faeghe Ghasemi.

**Writing – original draft:** Michael E. Symonds, Mohammad Hossein Sakhaei.

**Writing – review & editing:** Mousa Khalafi, Michael E. Symonds.

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
