## [Decision Letter · Decision Letter 0]

23 Aug 2023

PONE-D-23-23530The effects of exercise training on circulating adhesion molecules in adults: A systematic review and meta-analysisPLOS ONE

Dear Dr. Symonds,

Thank you for submitting your manuscript to PLOS ONE. After careful consideration, we feel that it has merit but does not fully meet PLOS ONE’s publication criteria as it currently stands. Therefore, we invite you to submit a revised version of the manuscript that addresses the points raised during the review process. Please submit your revised manuscript by Oct 07 2023 11:59PM. If you will need more time than this to complete your revisions, please reply to this message or contact the journal office at plosone@plos.org. Please include the following items when submitting your revised manuscript:A rebuttal letter that responds to each point raised by the academic editor and reviewer(s). You should upload this letter as a separate file labeled 'Response to Reviewers'.A marked-up copy of your manuscript that highlights changes made to the original version. You should upload this as a separate file labeled 'Revised Manuscript with Track Changes'.An unmarked version of your revised paper without tracked changes. You should upload this as a separate file labeled 'Manuscript'.

We look forward to receiving your revised manuscript.

Kind regards,

Ricardo Ney Oliveira Cobucci, Ph.D

Academic Editor

PLOS ONE

Journal Requirements:

2. Please include a copy of Table 1 which you refer to in your text on page 11.

Additional Editor Comments:

Dear authors,

as you can see, the reviewers have requested substantial revisions to your manuscript. We are certainly willing to reconsider a revised submission, but please know that this is not preliminary acceptance of your paper. When returning your revised manuscript, please be sure to include a point-by-point summary of the suggestions of the reviewers that specifies how and where in the text you have addressed the suggestions.

Reviewers' comments:

Reviewer's Responses to Questions

**Comments to the Author**

1. Is the manuscript technically sound, and do the data support the conclusions?

Reviewer #1: Yes

Reviewer #2: Yes

2. Has the statistical analysis been performed appropriately and rigorously? 

Reviewer #1: Yes

Reviewer #2: Yes

3. Have the authors made all data underlying the findings in their manuscript fully available?

Reviewer #1: Yes

Reviewer #2: Yes

4. Is the manuscript presented in an intelligible fashion and written in standard English?

Reviewer #1: Yes

Reviewer #2: Yes

5. Review Comments to the Author

Reviewer #1: I am glad for the opportunity to review this manuscript. I hope some questions and comments below contribute to the paper.

#1 - Abstract - Methods - Language restrictions and the last search date should be stated.

#2 - Abstract - Results/Conclusion - The authors should provide a general interpretation of the study findings and a brief summary of the limitations and significant implications of the evidence included in the review.

#3 - Abstract - The protocol registered name, registration number, and where it can be accessed (e.g., Web address) should be provided in the Abstract.

#4 - Methods - Last search date should be stated.

#5 - Methods - Why did the authors not include other databases like Embase and CENTRAL? EMBASE used to be a relevant database for scoping/systematic reviews.

#6 - Methods - Did the authors not consider a gray literature search, such as Google Scholar, relevant to this study question?

#7 - Methods - Why did the authors include only articles in English?

#8 - Methods - The authors should consider showing the complete search strategies for all databases, including any filters and limits used, in the Supplementary Material.

#9 - Methods - Were reference lists examined? If yes, the authors should specify the types of references examined.

#10 - Methods - What were the exclusion criteria? The inclusion and exclusion criteria should be explained in more detail using the study characteristics (e.g., participants, setting, target condition(s), and study design) and report characteristics (e.g., years considered, language, and publication status), preferentially showing the components of the PICO framework or one of its variant. A table summarizing the inclusion and exclusion criteria should be present.

#11 - Methods - Were unpublished manuscripts and conference abstracts eligible for inclusion?

#12 - Methods - The study selection process should describe in more detail in the text: duplicates, title, and abstract screening, and full-text articles assessment for relevance. Did the authors use a reference management software package or a screening and data extraction tool to study selection?

#13 - Methods - All independent variables and outcomes should be explicitly stated.

#14 - Methods - The authors used the Physiotherapy Evidence Database (PEDro) tool to assess the risk of bias of the included studies in the review. Did the authors consider adding an assessment to evaluate the quality of the evidence in the review, such as the GRADE framework?

#15 - Discussion - Thea authors should discuss limitations at the study and outcome level (e.g., risk of bias) and the review level (e.g., incomplete retrieval of identified research, reporting bias).

#16 - Conclusion - The conclusion is too short. The critical implications of the evidence included in the review should be stated.

Reviewer #2: The manuscript provides a solid basis for supporting the authors' findings, but an updated search is recommended. To evaluate the results, it is necessary to classify the quality of evidence using the GRADE method. It is necessary to ensure clear and legible resolutions for all figures and tables to avoid ambiguity.

6. PLOS authors have the option to publish the peer review history of their article (what does this mean?). If published, this will include your full peer review and any attached files.

Reviewer #1: No

Reviewer #2: No

---

## [Author Response · Author response to Decision Letter 0]

14 Sep 2023

Review Comments to the Author

Reviewer #1:

Comment: #1 - Abstract - Methods - Language restrictions and the last search date should be stated.

Response: Amended as requested.

Comment: #2 - Abstract - Results/Conclusion - The authors should provide a general interpretation of the study findings and a brief summary of the limitations and significant implications of the evidence included in the review.

Response: Amended as requested.

Comment: #3 - Abstract - The protocol registered name, registration number, and where it can be accessed (e.g., Web address) should be provided in the Abstract.

Response: Amended as requested.

Comment: #4 - Methods - Last search date should be stated.

Response: Amended as requested.

Comment: #5 - Methods - Why did the authors not include other databases like Embase and CENTRAL? EMBASE used to be a relevant database for scoping/systematic reviews.

Response: Amended as requested. An electronic search was conducted in PubMed, Scopus, Web of Science and Embase databases and search were updated until 31 August 2023. Please see Method- Search strategy section.

Comment: #6 - Methods - Did the authors not consider a gray literature search, such as Google Scholar, relevant to this study question?

Response: The references list of selected studies and Google Scholar were manually searched to ensure that all relevant studies were included in the meta-analysis. Please see Method- Search strategy section.

Comment: #7 - Methods - Why did the authors include only articles in English?

Response: Limiting systematic reviews to English-only is common in systematic reviews and articles published in non-English languages require translation, which may often be accompanied by errors.

Comment: #8 - Methods - The authors should consider showing the complete search strategies for all databases, including any filters and limits used, in the Supplementary Material.

Response: Amended as requested. Please see supplementary table 1.

Comment: #9 - Methods - Were reference lists examined? If yes, the authors should specify the types of references examined.

Response: The references list of all selected studies and Google Scholar were manually searched to ensure that all relevant studies were included in the meta-analysis. Please see Method- Search strategy section. 

#10 - Methods - What were the exclusion criteria? The inclusion and exclusion criteria should be explained in more detail using the study characteristics (e.g., participants, setting, target condition(s), and study design) and report characteristics (e.g., years considered, language, and publication status), preferentially showing the components of the PICO framework or one of its variant. A table summarizing the inclusion and exclusion criteria should be present.

Response: Added as requested. Please see Table 1.

#11 - Methods - Were unpublished manuscripts and conference abstracts eligible for inclusion?

Response: Unpublished manuscripts and conference abstracts were excluded. 

#12 - Methods - The study selection process should describe in more detail in the text: duplicates, title, and abstract screening, and full-text articles assessment for relevance. Did the authors use a reference management software package or a screening and data extraction tool to study selection?

Response: Added as requested. Please see Method- Eligibility Criteria and study selection.

#13 - Methods - All independent variables and outcomes should be explicitly stated.

Response: Added as requested. Please see Method- Eligibility Criteria and study selection.

#14 - Methods - The authors used the Physiotherapy Evidence Database (PEDro) tool to assess the risk of bias of the included studies in the review. Did the authors consider adding an assessment to evaluate the quality of the evidence in the review, such as the GRADE framework?

Response: Added as requested. 

Please see Method- Quality assessment and sensitivity analyses selection, Results- Quality assessment and sensitivity of analyses section and supplementary Table 3. 

#15 - Discussion - Thea authors should discuss limitations at the study and outcome level (e.g., risk of bias) and the review level (e.g., incomplete retrieval of identified research, reporting bias).

Response: Added as requested. Please see Discussion – Limitation section. 

#16 - Conclusion - The conclusion is too short. The critical implications of the evidence included in the review should be stated.

Response: Added as requested. Please see Discussion – Limitation section. 

Reviewer #2:

Comment: The manuscript provides a solid basis for supporting the authors' findings, but an updated search is recommended. To evaluate the results, it is necessary to classify the quality of evidence using the GRADE method. It is necessary to ensure clear and legible resolutions for all figures and tables to avoid ambiguity.

Response: Amended as requested. An electronic search was conducted in PubMed, Scopus, Web of Science and Embase databases and search were updated until 31 August 2023. Please see Method- Search strategy section. 

For GRADE, Please see Method- Quality assessment and sensitivity analyses selection, Results- Quality assessment and sensitivity of analyses section and supplementary Table 3, and Figures.

---

## [Decision Letter · Decision Letter 1]

27 Sep 2023

The effects of exercise training on circulating adhesion molecules in adults: A systematic review and meta-analysis

PONE-D-23-23530R1

Dear Dr. Symonds,

We’re pleased to inform you that your manuscript has been judged scientifically suitable for publication and will be formally accepted for publication once it meets all outstanding technical requirements.

Kind regards,

Ricardo Ney Oliveira Cobucci, Ph.D

Academic Editor

PLOS ONE

Additional Editor Comments (optional):

Congratulations!

Reviewers' comments:

Reviewer's Responses to Questions

**Comments to the Author**

1. If the authors have adequately addressed your comments raised in a previous round of review and you feel that this manuscript is now acceptable for publication, you may indicate that here to bypass the “Comments to the Author” section, enter your conflict of interest statement in the “Confidential to Editor” section, and submit your "Accept" recommendation.

Reviewer #1: All comments have been addressed

Reviewer #2: All comments have been addressed

2. Is the manuscript technically sound, and do the data support the conclusions?

Reviewer #1: Yes

Reviewer #2: Yes

3. Has the statistical analysis been performed appropriately and rigorously? 

Reviewer #1: Yes

Reviewer #2: Yes

4. Have the authors made all data underlying the findings in their manuscript fully available?

Reviewer #1: Yes

Reviewer #2: Yes

5. Is the manuscript presented in an intelligible fashion and written in standard English?

Reviewer #1: Yes

Reviewer #2: Yes

6. Review Comments to the Author

Reviewer #1: The authors have made appropriate adjustments to the original submission.

All my comments have been answered, and I have no further recommendations.

Reviewer #2: (No Response)

7. PLOS authors have the option to publish the peer review history of their article (what does this mean?). If published, this will include your full peer review and any attached files.

Reviewer #1: No

Reviewer #2: No

---

## [Editor Report · Acceptance letter]

5 Oct 2023

PONE-D-23-23530R1 

The effects of exercise training on circulating adhesion molecules in adults: A systematic review and meta-analysis 

Dear Dr. Symonds:

I'm pleased to inform you that your manuscript has been deemed suitable for publication in PLOS ONE. Congratulations! Your manuscript is now with our production department. 

Kind regards, 

on behalf of

PROFESSOR Ricardo Ney Oliveira Cobucci 

Academic Editor

PLOS ONE